# The Role of "Nostalgia" in Environmental Restorative Effects from the Perspective of Healthy Aging: Taking Changchun Parks as an Example

**Tianjiao Yan [1,2], Hong Leng [1,2,*] and Qing Yuan [1,2]**

1 School of Architecture, Harbin Institute of Technology, Harbin 150001, China; ytjhit@126.com (T.Y.); hityq@126.com (Q.Y.)
2 Key Laboratory of National Territory and Spatial Planning and Ecological Restoration in Cold Regions, Harbin 150001, China
* Correspondence: hitlaura@126.com

**Abstract:** Aging and elderly health issues have always been the focus of attention, both within and outside the industry. With the introduction of the national "14th Five-Year Plan" for healthy aging, it is urgent to address how to implement this plan. Among them, the restorative environment is an important part of implementing healthy aging. For older adults, "nostalgia" is a common emotional experience, and "nostalgia therapy" is also commonly used for mental health recovery, which has important significance for healthy aging. However, although existing research on "nostalgia" has already involved local attachment and the environment, there are few studies that use space as a carrier in the context of environmental restorative effects. Therefore, from the perspective of healthy aging, combined with structural equation modeling, this study took four parks in Changchun City as examples to explore the role of "nostalgia" in the restorative effect of the park environment. It found that, firstly, both the "nostalgia inclination" influenced by individual conditions and the "landscape perception" influenced by landscape quality had a positive impact on the "nostalgia affection"; secondly, nostalgia affection and place attachment were important mediating factors for environmental restorative effects, and the pathways of "landscape perception → nostalgic affection → environmental restorative effects", "landscape perception → place attachment → environmental restorative effects", and "landscape perception → nostalgic affection → place attachment → environmental restorative effects" all existed. Based on the above path exploration, corresponding spatial optimization ideas for effectively improving the health level of older adults have been provided.

**Keywords:** nostalgia; environmental restorative effects; healthy aging; landscape perception; place attachment

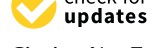



## 1. Introduction

According to public information from the National Bureau of Statistics of China, as of 2022, the population aged 60 and above is approximately 280 million people, accounting for 19.8%, of which 210 million are aged 65 and above, accounting for 14.9%. Internationally, it is generally believed that when a country or region has 10% of the population aged 60 and above or 7% of the population aged 65 and above, it means that the population of the country or region is in an aging society. It can be seen that China has already entered an aging society. Among them, the Northeast region is one of the regions with the deepest degree of aging in China, with over 23% of the elderly population above 60, represented by cities such as Shenyang and Changchun. The large base of China's aging population is often accompanied by elderly health issues [1], especially mental health issues. For example, changes in family structure, the occurrence of loss events, and the decline of social roles make "empty nest", widows or major physical illnesses, retired older adults, and others more prone to mental health problems [2,3]. To address the issue of

population aging, the concept of "healthy aging" was proposed [4]. In 1987, the World Health Assembly expanded its concept to include biological aging and social aging, with its core including physical health, mental health, and good social adaptation. In the 2016 World Health Organization's "Global Aging and Health Report" [5] and the "14th Five Year Plan for Healthy Aging" jointly issued by 15 departments including the National Health Commission of China in 2022, the focus was shifted from "diseases" to "functions", emphasizing the need to optimize the social environment for healthy living of the elderly, better meet their health needs, improve their health level, and extend their expected lifespan and quality. Overall, healthy aging mainly depends on the intrinsic abilities, supportive environment, and interactions of older adults [6]. The supportive environment in cities mainly includes the configuration of various infrastructures and the provision of green space. Most studies have considered the quantity, quality, and accessibility of facilities and spaces, but there are still many urban renewal and renovation projects that are costly but have little effect. The fundamental reason is that they do not understand the true needs of older adults and cannot implement the goal of healthy aging.

The restorative environment is an important health-supporting environment in cities. According to Attention Restoration Theory (ART) [7,8] and Stress Reduction Theory (SRT) [9], restorative environments refer to environments that help residents recover from mental fatigue and negative emotions. The former indicates that as long as people take a look at an environment with natural elements (such as trees, flowers, water, etc.), even in a few minutes, their attention can be restored because natural contact can allow the brain to eliminate functions that interfere with concentration and obtain necessary rest and recovery [10]. The latter indicates that in the long-term evolutionary adaptation process, the natural living environment affects humans, making it easier for people to resonate with the natural environment. The more natural the characteristics of the environment they are in, the more likely they are to develop positive emotions and reduce negative emotions or stress [11]. Among them, parks, as important restorative environments in cities, have long been recognized by the industry and the public for their physical and mental health benefits. At the same time, parks are the most common natural space chosen by older adults, and the optimization of this environment should be an important part of healthy aging. Some studies have revealed that older adults who live near parks and have good interaction with parks have better psychological well-being [12]. Additionally, park environments can improve energy levels, restore attention, alleviate stress, improve mood, and enhance happiness through the provision of good spatial elements and spaces with different attributes [13–16]. At present, relevant research focuses on exploring the correlation between park characteristics and the health recovery effects of specific populations. For instance, greener and more convenient spaces are more conducive to mental health recovery, which leads to the blind addition of plants and facilities to park planning or renovation plans, exacerbating the homogenization problem of the park without touching on the intrinsic emotional needs of special groups.

Some studies have established the relationship between park environment preference, place attachment, and environmental restoration effects [17,18]. Environmental preference is an attitude, while place attachment is an emotional response. Currently, parks not only serve as functional carriers for residents' activities but also fulfill their emotional needs, especially for older adults. Due to differences in cognitive processes, older adults have developed unique emotional needs towards the environment. For example, older adults tend to prefer familiar environments [19]. They also experience stronger emotional responses triggered by specific contexts, and nostalgia is more common among the elderly population [20–22]. Nostalgia is considered a form of self-awareness and positive emotion in modern psychology research and originates from beautiful memories and yearning for childhood, intimate relationships, or atypical positive events [23]. It has the functions of storing positive emotions, enhancing self-positivity, strengthening social connections, increasing a sense of belonging, providing meaning in life, unifying the self, and adapting to life [24]. Studies have pointed out that triggering nostalgia in older adults, through recalling

and experiencing past experiences and environments, can enhance their sense of attachment to the environment, alleviate feelings of loss, and provide great emotional satisfaction. This leads to better psychological well-being and subjective happiness [25,26]. Nostalgia therapy is an important way to improve the mental health of older adults, and landscape therapy is an important way to trigger nostalgia, which has been applied in medical environments [27]. The research on urban open spaces—parks also advocates for "healing landscapes" [28] and "green prescriptions" [29] from the perspective of health restoration. Some studies have also verified the relationship between landscape aesthetics, nostalgia, and a sense of locality [30,31]. Many natural landscapes can promote people's positive reflection, identify or reconstruct their sense of environmental identity and belonging, and thus play a "healing" role [32–34]. However, existing research on space and nostalgia has focused on tourism [35–37], with few studies focusing on urban green space and health goals. As a consequence, discussing the park environment from an emotional perspective is of great significance for both improving urban landscapes and enhancing public health levels. More importantly, parks have become a way of life for older adults, making it even more meaningful to explore how parks can enhance their health restoration through emotional fulfillment.

In summary, there are two major issues with existing research: First and foremost, the current research findings on the restorative environment of parks have led to a significant homogenization of parks. A park without distinctive features and recognition can lead to feelings of "nowhere to go" and "tasteless food" among older adults. Meanwhile, some theme parks or art parks in China that have been heavily invested in creating them are facing the embarrassment of being "barren gardens". This reflects the lack of in-depth exploration of cultural and special emotional needs. Secondarily, the nostalgia commonly seen in older adults has not been taken seriously in research on place attachment and environmental restorative effects. Nostalgia requires the support of space and context, and there are few existing studies that intervene from the perspectives of healthy aging and spatial planning, which cannot guide the design and renovation of parks. Therefore, this study was based on the background of healthy aging, using the park environment as a spatial carrier and innovatively introducing the "nostalgia" affection into the restorative environment. With this study, we aim to specifically understand how nostalgia, as a key emotion experienced by older adults, plays a role in the restorative effects of parks and how this can inform the planning and design of more effective and health-promoting spaces.

## 2. Research Hypothesis

The restoration of the environment is achieved through complex interactions between individuals and contexts. Based on existing research findings, this study decomposed the psychological process and added the mediating role of nostalgia (Figure 1). The overall research hypotheses include three parts:

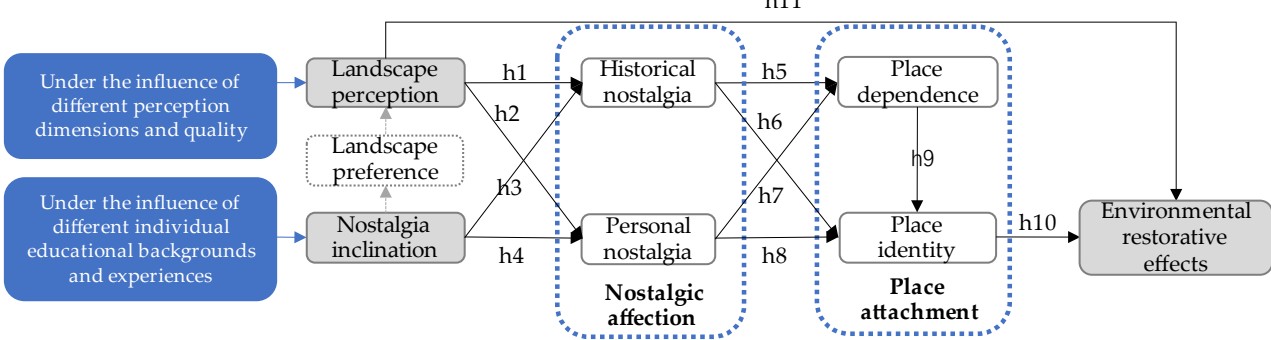

**Figure 1.** Research hypothesis.

Firstly, the phenomenon of nostalgia is common in the elderly population in urban areas, which is related to the psychological development stage of older adults. Studies

in psychology and psychiatry have used nostalgia therapy to help older adults regain familiar life experiences, reduce loneliness and feelings of loss, improve their cognitive and emotional experiences, enhance their sense of happiness, and prevent and improve mild cognitive impairment [38]. Nostalgia is formed by the common experiences, memories, and historical customs of older adults and can be discussed in two dimensions: nostalgic inclination and nostalgic affection. Nostalgic inclination refers to the attitudes held by individuals towards the past, influenced by their own upbringing and other factors; nostalgic affection refers to the emotional experience in the environment, which is also a preference and can be further divided into personal nostalgia and historical nostalgia [39]. Personal nostalgia intimates the direct experiences of individuals, while historical nostalgia means collective experiences. As proof, most elderly people have lived in rural areas, and landscapes with rural characteristics may evoke their memories of certain moments or events in the past, thereby influencing their overall feedback on the environment. For that reason, nostalgia has been explored in rural tourism [40,41]. Meanwhile, sensory dimensions and quality also influence landscape perception and play an important role in triggering nostalgia. Baker and Kennedy believed that nostalgic emotions require contextual stimuli to occur [42]. Stimuli that act on different senses in a tourist context can induce nostalgic feelings in people [43], such as food, music, and smells that are perceived by the senses [44–46].

As a result, it proposed the following hypotheses regarding landscape perception, nostalgic inclination, and nostalgic affection: h1–h4 (Table 1). These assumptions can help us understand the composition and triggering mechanism of "nostalgia", which is an important basis for this study.

Secondly, place attachment originates from the fields of geography and psychology, referring to the connection between individuals and specific environments, often measured by two dimensions: place identity and place dependence [47]. Environmental psychologists also believe that affection plays a central role in the connection between people and places. Place dependence refers to people's functional attachment to specific places, such as specific environmental features and attributes of places that meet specific activity needs of tourists, representing the unique qualities of a place [48]; place identity highlights the symbolic significance of a place and is also considered as individuals' strong emotional attachment to specific locations or environments, as well as the connection between a location and individual identity, which includes cognitive and affective factors [49,50]. Some studies have also shown that place attachment mainly affects people's attitudes and behaviors towards the environment through the mediating role of place identity [51]. In previous research on nostalgia tourism, the relationship between nostalgia and tourism perception and behavior has been established, and place attachment has also been recognized as an important variable [52].

Consequently, h5–h9 hypotheses were proposed regarding nostalgia and place attachment (Table 1). These assumptions are an important component of this research and help people understand the mechanism of "nostalgia".

Thirdly, there are two psychological paths related to environmental preference and restorative effect: a simple influence path of "environmental preference → environmental restorative effects" and a comprehensive complex influence path of "environmental preference place dependence → place identity → environmental restorative effects" [17], both of which have been confirmed. The higher the environmental preference, the higher the place attachment, and the better the restorative effect of the environment on individuals [53–55].

On the basis of the above evidence, h10–h11 can be proposed (Table 1). These two hypotheses are the basis for this research, both aimed at improving environmental resilience. In addition, it is necessary to verify whether nostalgic emotion and place attachment are mediating factors influencing the restorative effects of the environment through landscape perception.

**Table 1.** Assumption content and justification.

| Involving Dimensions | Assumption Content | Justification |
|---|---|---|
| landscape perception, nostalgic inclination, and nostalgic affection | h1. Landscape perception can positively influence historical nostalgia | Some landscape may evoke memories of past times for individuals, such as elderly people living in rural environments who may resonate with rural landscape and evoke nostalgia |
| | h2. Landscape perception can positively influence personal nostalgia | Certain scenes can evoke good memories of an individual's childhood and the past, for example, places where children often play have iconic trees or swings |
| | h3. Nostalgic inclination can positively influence historical nostalgia | Individuals who hold a positive attitude towards memories may be awakened by the scene to recall past times or memories |
| | h4. Nostalgic inclination can positively influence personal nostalgia | |
| nostalgic affection and place attachment | h5. Historical nostalgia can positively influence place dependence | If the scene satisfies an individual's longing for the past, it may enhance their functional dependence and emotional identity towards the place |
| | h6. Historical nostalgia can positively influence place identity | |
| | h7. Personal nostalgia can positively influence place dependence | If the scene satisfies an individual's positive memories of their past, it may enhance their functional dependence and emotional identity towards the place, such as a sense of belonging to "home" |
| | h8. Personal nostalgia can positively influence place identity | |
| | h9. Place dependence can positively influence place identity | Places that meet basic functional dependencies may form an individual's sense of belonging and identification with them |
| landscape perception, place identity, environmental restorative effects | h10. Place identity can positively influence the restorative effects of the environment | Good visual, auditory, and olfactory landscapes, or familiar and belonging landscapes, may help individuals stay away from stress, feel energized, and improve their emotional state |
| | h11. Landscape perception can positively influence the restorative effects of the environment | |

## 3. Materials and Methods

### 3.1. Scale Design

The main measurement components of the survey include 5 dimensions and 7 latent variables, namely landscape perception, nostalgia inclination, individual nostalgia, historical nostalgia, place attachment, place identification, and environmental restorative effects (Table 2). All of them were rated on a Likert scale from 1 to 5. Landscape perception was an interactive process between the landscape environment and people, and it was also an individual's overall subjective evaluation of the landscape. The evaluation of landscape perception drew on the division of landscape elements and sensory dimensions [56], incorporating olfactory perception [57]. It included 8 factors: L1 visual beautification of landscape topography, L2 guidance of landscape roads [58], L3 affective response to landscape sculptures [59], L4 color configuration of plant landscape [60], L5 distribution of landscape facilities, L6 cultural characteristics of symbolic landscape [61], L7 affective enhancement of soundscape (good artificial and natural sounds such as music playback, birds singing, and flowing water, without obvious noise interference) [62], and L8 atmospheric rendering of olfactory landscape (good natural scent, such as floral aroma, with no obvious adverse odor interference) [63]. In this study, "nostalgia" was divided into two dimensions: nostalgia inclination and nostalgia affection. The former referred to the individual's own attitude towards nostalgia, while the latter referred to the level of nostalgia emotion stim-

ulation in the environment. The nostalgia inclination items were predicated on scales by Holbrook and Schindler [64], consisting of three items. The nostalgia affection scale was in accordance with the two-dimensional division proposed by Chris Marchegiani and Ian Phau [65], with two items for historical nostalgia and individual nostalgia separately. Place attachment represents an evaluation of a person's emotional connection to the environment. The place attachment scale stemmed from Williams et al.'s [47] division of place dependence and place identity, each containing three items. The environmental restoration effects were the subjective evaluation of the health restoration potential of the environment by older adults. Due to the aforementioned mental health issues in older adults and the emotional context of nostalgia, this study focused mainly on the restoration of the mental health dimension. The measurement was combined with the ROS [7], WHO-5 [66], and POMS [67] scales to examine the dimensions of pleasure, vitality, fatigue, and focus most commonly involved in the mental health of older adults [68], forming four items. Moreover, it investigated landscape preferences, which were divided into natural landscape, traditional architecture, plant landscape, humanistic landscape, and rural landscape based on existing classifications [69,70]. Among them, the cultural landscape refers to landscapes with symbolic value under historical or cultural influence [71]. Besides its traditional significance, the rural landscape here also refers to small pieces or landscapes with rural characteristics and qualities [72].

### 3.2. Data Collection

Changchun City, Jilin Province, a typical city in Northeast China, was selected as the survey site. There are three reasons for choosing Changchun: first, as of 2021, the population aged 60 and above accounted for 20.85% of the city's population, while the population aged 65 and above accounted for 14.15%, making the aging problem very severe. Secondly, Changchun was one of the earliest cities in China to propose the construction and planning of a forest city. As of the end of 2020, the green coverage rate of the built-up areas was 41.12%, making it a national "garden city" and "forest city". Thirdly, Changchun has city name cards for "Forest City", "Automobile City", and "Film City", with a rich cultural background. Therefore, Changchun is the preferred research location for this study. Furthermore, Nanhu Park, Changchun Park, Shengli Park, and Laodong Park were selected as sample parks, all of which are representative, historical, and open parks in Changchun (Table 3). All four sample parks have water features, well-equipped facilities, landscape sculptures, and symbolic landscapes. Nanhu Park and Laodong Park mainly feature natural soundscapes, while Shengli Park and Changchun Park are complemented by artificial soundscapes such as music and recorded bird songs. The four all have flowers, especially Changchun Park and Nanhu Park, which have better-scented landscapes. In addition to the park's own characteristics, the preference space of older adults in various parks has also been examined. The numbers of older adults were higher, and their stay times were longer in the waterfront space, vegetation space, and garden road space in these four parks.

This research selected sunny days in the summer. The process was as follows: Firstly, the researchers recruited elderly participants at the entrance and exit of the sample parks, and the screening criteria were: (1) older adults aged 55 and above without intellectual disabilities. (2) Participants were not the first to visit the sample park and had a certain level of familiarity with the park they were visiting. This was because research has shown that familiarity has an impact on emotional perception [16], and there was a survey on local attachment in this study. First-time visitors are usually unable to answer their level of dependence on the environment, and restorative effects may also be mediated by freshness. Therefore, to avoid bias in the results, it excluded older adults who entered the park for the first time. The researchers then explained the research objectives to the participants, read the questionnaire content, and recorded the results. Finally, they distributed small gifts and obtained personal information.

**Table 2.** Scale dimensions and item design.

| Variable Dimension | | Question Items | Item Quantity | Source |
|---|---|---|---|---|
| F1 landscape perception | | L1 visual beautification of landscape topography; L2 guidance of landscape roads; L3 affective response to landscape sculptures; L4 color configuration of plant landscape; L5 distribution of landscape facilities; L6 cultural characteristics of symbolic landscape; L7 affective enhancement of soundscape; and L8 atmospheric rendering of olfactory landscape | 8 | expert discussion, [56–63] |
| F2 nostalgia inclination | | N1 I miss the place where I used to live; N2 I still miss the things I ate in my childhood; N3 I still like the songs I heard when I was a child | 3 | [64] |
| nostalgic affection | F3 historical nostalgia | H1 Here gives me a positive feeling about the past era; H2 Here makes me feel integrated into my previous life | 2 | [65] |
| | F4 personal nostalgia | P1 Here has awakened my memories of the good old times; P2 Sometimes I want to go back to the simple life of the past | 2 | |
| place attachment | F5 place dependence | D1 Here is the most suitable place for me to relax and relax; D2 Recreation here is more satisfying than other parks; D3 The leisure experience here is irreplaceable by other parks | 3 | [47,73] |
| | F6 place identity | I1 I really agree with the environment here; I2 Here has special significance for me; I3 Here makes me linger and forget to return | 3 | |
| F7 environmental restorative effects | | R1 Eliminate fatigue (I feel like I'm away from trivial life, and feel comfortable and relaxed); R2 Energize (I feel full of vitality); R3 Improve emotions (I feel happy and peaceful); R4 Restore attention (I feel clear, focused, and alert) | 4 | [7,66] |
| landscape preference | | natural landscape; traditional architecture; plant landscape; humanistic landscape (such as exhibition hall); rural landscape | | [67–72] |

A pilot survey was conducted in June 2022, resulting in 70 questionnaires. The landscape perception part is the overall subjective perception evaluation of older adults in the parks. The results showed that older adults had a lower perception of the distribution of indicator guidance systems and facilities, and the load coefficient was lower in this dimension. According to this, adjustments were made to the observed variables, and the items related to L2 guidance of landscape roads and L5 distribution of landscape facilities were removed. Proceeding to the next step, a formal survey was conducted in August 2022, where a total of 210 valid questionnaires were collected, with an effective response rate of 84.2%. 60, 55, 53, and 37 questionnaires were collected in Nanhu Park, Changchun Park, Shengli Park, and Labor Park, respectively. The questionnaires consisted of two parts: the personal information of the elderly participants and the measurement section. The personal information included gender, age, education level, and rural residence experience. Female elderly participants accounted for 58.57% of the sample. Regarding age, 10% were between

55 and 59 years old, 38.57% were between 60 and 69 years old, 47.14% were between 70 and 79 years old, and 4.29% were 80 years old and older. 62.86% had a primary school education or below, and 82.86% had rural residence experience. The measurement section included a survey of the seven latent variables and landscape preferences mentioned above.

**Table 3.** Information of sample parks.

| Park Name | Nanhu Park | Changchun Park | Shengli Park | Laodong Park |
| --- | --- | --- | --- | --- |
| Area | 238.6 ha | 66 ha | 24.5 ha | 16.5 ha |
| Construction year | 1935 | 1999 | 1915 | 1958 |
| Characteristics | city level comprehensive park; city name card; beautiful lake scenery | ecological park; relying on a combination of ornamental plants and garden buildings | long history; combining Japanese garden style | typical classical architectural style in northern China; with numerous antique buildings |
| On-site photos |  |  |  |  |

### 3.3. Statistical Methods

The structural equation model (SEM) was selected for model fitting and mediating effect testing. SEM is a multivariate statistical model that combines structural and measurement models, breaking the barriers of traditional regression methods and being able to handle complex relationships among multiple variables [74]. It has been widely applied in the social sciences [75]. Especially for this study, it is not suitable for older adults to use highly stimulating pressure experiments, and this survey content is related to emotions and subjective feelings. Thus, evaluating the evaluation results in the form of a scale is the most suitable way for older adults. In addition, this survey involves different dimensions and processes of action, SEM is the most suitable method for this study. Initially, the data was imported into SPSS 22.0, and the measurement model was estimated and tested through validation factor analysis, etc. Reliability and validity analysis of data is a prerequisite for everything, and factor analysis can only be conducted by passing the overall and various dimensions of indicators. Furthermore, structural validity, aggregated validity, and discriminant validity tests need to be conducted. If the load coefficient is >0.40, it indicates that the observed variable can better explain the latent variable; if the AVE value is >0.50 and the CR > 0.70, it suggests that each factor has a good concentration in this dimension; if the AVE square root is greater than the correlation coefficient value, it illustrates that the degree of difference between different factors is significant. After adjusting the model to meet the above indicators, path analysis was conducted using AMOS 22.0 to estimate and test the structural model. What is more, an independent sample *t*-test, a one-way ANOVA test, and Pearson analysis were combined to discuss the results of different dimensions.

## 4. Results

### 4.1. Evaluation Results of Various Dimensions

Scores of 1–5 indicate strong disagreement, disagreement, neutrality, agreement, and strong agreement, respectively. First and foremost, in the evaluation of environmental restorative effects, the average score of R1–R4 ranged from 4.11 to 4.34, both of which belong to agree (4) to very agree (5), indicating that the health restorative effect of the sample park on older adults was positive and beyond doubt. This was consistent with the results of a multi-sensory dimension study in Guangzhou, as well as the consensus within and outside the industry [76]. Among them, the ranking of these four scores is: R3 improves emotions, R2 energizes, R1 eliminates fatigue, and R4 restores attention. Comparing the landscape perception evaluations of the four sample parks, it can be seen that due to the abundance of flowers in Nanhu Park and Changchun Park and the good maintenance and cutting of plants in Shengli Park, older adults in these three parks showed a stronger preference and satisfaction towards colorful botanical gardens, lawns, and tree shade (L4). Due to the long history, unique exhibition halls, and rustic landscape sketches of Shengli Park and the distinctive structures of Laodong Park, elderly participants had a higher subjective rating for landscape sculptures (L3) and symbolic landscapes (L6). Furthermore, in the evaluation of landscape perception, it was between neutral (3) and strongly agreed (5), with L4 (4.14) and L3 (4.08) having the highest average score and L1 (3.75) and L8 (3.71) having the lowest. This illustrated that older adults in the sample parks had the strongest perception of plants and sculptures and a weaker perception of terrain and odorous landscapes. Furthermore, in the dimensions of nostalgia and place attachment, the average scores of nostalgia inclination I1–I3 were between 3.66 and 3.89, and the four items of nostalgia emotion were between 3.63 and 3.83, all between neutrality (3) and agreement (4), indicating that most older adults have nostalgia inclination and affection. The average scores of place dependence and place identity were 3.93 and 4.09, respectively, illustrating that older adults have attachment attitudes towards the sample parks.

The above results partially confirmed the subjective attitudes of older adults in these parks, such as their confirmation that the park environment can have a restorative effect, as well as the presence of "nostalgia" and "a sense of place", which provided a basis for further analysis.

### 4.2. Reliability and Validity Testing

The tests of reliability and validity were conducted initially. The overall Cronbach $\alpha$ was 0.938, and the KMO from Bartlett's test of sphericity was 0.803. The reliability and validity of each dimension passed the tests (Table 4). The discriminant validity also passed the test (Table 5), indicating that the model has achieved the required level of rationality and usability. Subsequently, the study performed model fitting with $\chi2/df = 1.816$, and the indices such as GFI = 0.748 > 0.7, RMSEA = 0.093 < 0.1, RMR = 0.036 < 0.05, CFI = 0.930 > 0.7, NFI = 0.859 > 0.7, and NNFI = 0.911 > 0.7 met the acceptable standards [76,77].

The results of this section were the prerequisite for further discussion on the existence of the paths. Overall, all indicators of the model have passed the test and can be further analyzed.

### 4.3. Model Fitting Results

Using structural equation modeling, the role of nostalgia in environmental restorative effects was obtained (Figure 2 and Table 6). Usually, if the *p*-value is less than 0.05, it is considered a hypothesis to be valid; if *p* is less than 0.01, it is more significant, and the ability to explain hypotheses is higher; In this study, the *p*-value is relaxed to 0.1, which means that the explanatory rate reaches 90%, and this hypothesis can be considered valid. The results indicated that all hypotheses were supported except for hypothesis 8. Landscape perception had significant positive impacts on historical nostalgia (0.405), personal nostalgia (0.615), and environmental restorative effects (0.433). Nostalgia inclination impacted nostalgia affection (0.447 and 0.352) significantly and positively; historical nostalgia had

significant positive effects on place dependence (0.409) and place identity (0.154), while personal nostalgia had a significant positive effect on place dependence (0.359); and, place dependence had an impact on place identity (0.872) and place identity on environmental restorative effects (0.336) significantly and positively. Furthermore, to proceed to the next step, bootstrap tests (sample = 5000) were conducted to examine the chain mediation effects of nostalgia and place attachment on the impact of landscape perception on environmental restorative effects (Tables 7 and 8). If the confidence interval of bootstrap does not include 0, it expresses the presence of a mediation effect. If the direct effect does not exist but the mediation effect does, it is considered complete mediation; otherwise, it is considered partial mediation [78]. The results implied that both nostalgia and place attachment were mediators in the environmental restorative effects, and paths with partial mediation, such as "landscape perception → nostalgia affection → environmental restorative effects" (0.140), "landscape perception → place attachment → environmental restorative effects" (0.108), and "landscape perception → nostalgia → place attachment → environmental restorative effects" (0.059), were supported.

**Table 4.** The reliability and validity test results of each dimension of the model.

| Latent Variables | Observational Variables | Std. Estimate | Cronbach $\alpha$ | AVE | CR |
|---|---|---|---|---|---|
| F1 landscape perception | L1 | 0.707 | 0.923 | 0.675 | 0.925 |
|  | L3 | 0.831 |  |  |  |
|  | L4 | 0.783 |  |  |  |
|  | L6 | 0.931 |  |  |  |
|  | L7 | 0.922 |  |  |  |
|  | L8 | 0.726 |  |  |  |
| F2 nostalgia inclination | N1 | 0.898 | 0.967 | 0.928 | 0.975 |
|  | N2 | 0.992 |  |  |  |
|  | N3 | 0.997 |  |  |  |
| F3 historical nostalgia | H1 | 0.961 | 0.970 | 0.949 | 0.974 |
|  | H2 | 0.988 |  |  |  |
| F4 personal nostalgia | P1 | 0.971 | 0.949 | 0.903 | 0.949 |
|  | P2 | 0.929 |  |  |  |
| F5 place dependence | D1 | 0.745 | 0.894 | 0.755 | 0.902 |
|  | D2 | 0.914 |  |  |  |
|  | D3 | 0.935 |  |  |  |
| F6 place identity | I1 | 0.881 | 0.901 | 0.769 | 0.909 |
|  | I2 | 0.865 |  |  |  |
|  | I3 | 0.885 |  |  |  |
| F7 environmental restorative effects | R1 | 0.853 | 0.946 | 0.815 | 0.946 |
|  | R2 | 0.974 |  |  |  |
|  | R3 | 0.951 |  |  |  |
|  | R4 | 0.825 |  |  |  |

**Table 5.** Differentiation validity test.

|  | F1 | F2 | F3 | F4 | F5 | F6 | F7 |
|---|---|---|---|---|---|---|---|
| F1 | **0.822** |  |  |  |  |  |  |
| F2 | 0.538 | **0.963** |  |  |  |  |  |
| F3 | 0.579 | 0.665 | **0.974** |  |  |  |  |
| F4 | 0.718 | 0.669 | 0.793 | **0.950** |  |  |  |
| F5 | 0.720 | 0.462 | 0.657 | 0.618 | **0.876** |  |  |
| F6 | 0.653 | 0.518 | 0.649 | 0.578 | 0.869 | **0.877** |  |
| F7 | 0.652 | 0.370 | 0.529 | 0.662 | 0.615 | 0.652 | **0.903** |

Note: Bold numbers represent the AVE square root value.

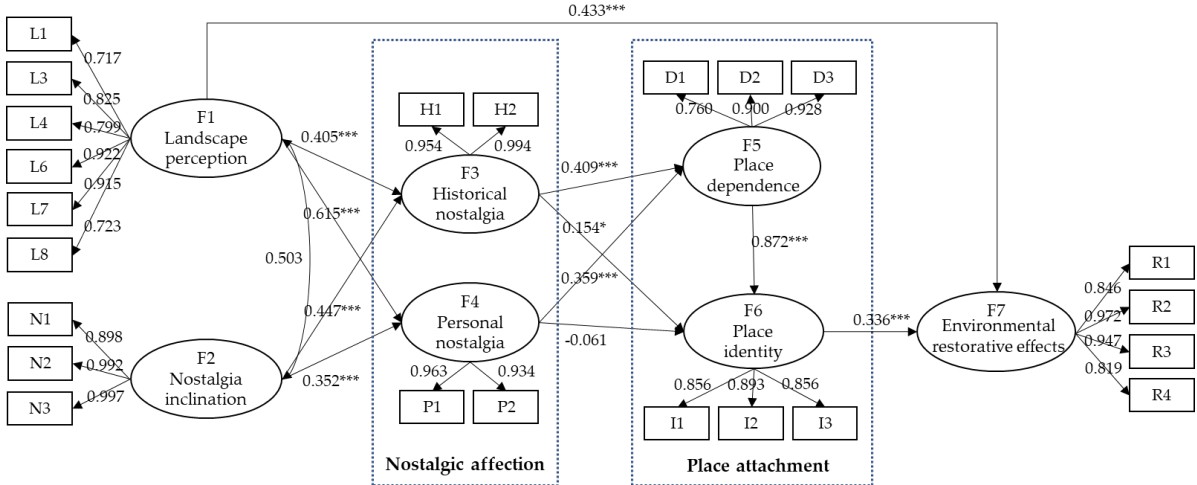

**Figure 2.** The role path and coefficient of nostalgia in environmental restoration effects (* *p* < 0.1, *** *p* < 0.01).

**Table 6.** The path results of nostalgia.

| Hypothesis | X | → | Y | Standardized Coefficient | SE | *t*-Value | *p*-Value | Hypothesis Testing |
|---|---|---|---|---|---|---|---|---|
| h1 | F1 | → | F3 | 0.405 | 0.133 | 3.645 | 0.000 *** | Established |
| h2 | F1 | → | F4 | 0.615 | 0.159 | 5.484 | 0.000 *** | Established |
| h3 | F2 | → | F3 | 0.447 | 0.071 | 4.381 | 0.000 *** | Established |
| h4 | F2 | → | F4 | 0.352 | 0.071 | 4.086 | 0.000 *** | Established |
| h5 | F3 | → | F5 | 0.409 | 0.097 | 3.156 | 0.002 *** | Established |
| h6 | F3 | → | F6 | 0.154 | 0.076 | 1.667 | 0.095 * | Established |
| h7 | F4 | → | F5 | 0.359 | 0.082 | 2.759 | 0.006 *** | Established |
| h8 | F4 | → | F6 | −0.061 | 0.064 | −0.662 | 0.508 | Not established |
| h9 | F5 | → | F6 | 0.872 | 0.156 | 6.108 | 0.000 *** | Established |
| h10 | F6 | → | F7 | 0.336 | 0.097 | 2.806 | 0.005 *** | Established |
| h11 | F1 | → | F7 | 0.433 | 0.102 | 3.358 | 0.001 *** | Established |

\* *p* < 0.1, *** *p* < 0.01.

**Table 7.** Effect process.

| | Items | Effect | SE | *t*-Value | *p*-Value | LLCI | ULCI |
|---|---|---|---|---|---|---|---|
| Direct effect | Landscape perception ⇒ Environmental restorative effects | 0.233 | 0.109 | 2.132 | 0.037 ** | 0.019 | 0.448 |
| | Landscape perception ⇒ Nostalgic affection | 0.844 | 0.108 | 7.827 | 0.000 *** | 0.633 | 1.055 |
| | Landscape perception ⇒ Local attachment | 0.567 | 0.134 | 4.225 | 0.000 *** | 0.304 | 0.830 |
| Indirect effect process | Nostalgic affection ⇒ Local attachment | 0.364 | 0.109 | 3.322 | 0.001 *** | 0.149 | 0.578 |
| | Nostalgic affection ⇒ Environmental restorative effects | 0.166 | 0.086 | 1.944 | 0.056 * | −0.001 | 0.334 |
| | Local attachment ⇒ Environmental restorative effects | 0.191 | 0.089 | 2.159 | 0.034 ** | 0.018 | 0.365 |
| Total effect | Landscape perception ⇒ Environmental restorative effects | 0.541 | 0.076 | 7.090 | 0.000 *** | 0.391 | 0.690 |

\* *p* < 0.1, ** *p* < 0.05, *** *p* < 0.01.

**Table 8.** Mediation effect test.

| Items | Effect | Boot SE | Boot-LLCI | Boot ULCI | Mediation Testing |
|---|---|---|---|---|---|
| Landscape perception ⇒ Nostalgic affection ⇒ Environmental restorative effects | 0.140 | 0.090 | 0.012 | 0.366 | Established |
| Landscape perception ⇒ Local attachment ⇒ Environmental restorative effects | 0.108 | 0.075 | 0.003 | 0.300 | Established |
| Landscape perception ⇒ Nostalgic affection ⇒ Local attachment ⇒ Environmental restorative effects | 0.059 | 0.042 | 0.002 | 0.164 | Established |

The results of this section clarified the assumptions made earlier and the role of nostalgia in the restorative effects of the park environment. In the mass, the results supported most hypotheses, and "nostalgia" did indeed play an important mediating role. The specific reasons and explanations will be discussed in detail below.

## 5. Discussion

*5.1. Discussion on the Decomposition Path of "Nostalgia" in the Environmental Restorative Effects*

The results of the model fitting indicated: at first, the paths from landscape perception (h1 and h2) and nostalgia inclination (h3 and h4) to nostalgic affection were established. Second, among the paths from nostalgic affection to place attachment (h5 to h9), the path from personal nostalgia to place identification (h8) was not established, while all other paths were established. Third, the path from local identity (h10) and landscape perception (h11) to environmental restorative effects was established.

(1) In the path from landscape perception to nostalgic affection, landscape perception had a positive impact on both historical nostalgia and personal nostalgia, with a larger path coefficient (0.615) for personal nostalgia. This was consistent with existing research findings that landscapes can evoke emotional resonance and memories of past times and events for the elderly through visual, auditory, and olfactory stimuli [79,80]. Landscape perception is a subjective evaluation influenced by environmental quality, and conveying positive landscape emotions can trigger individuals' pleasant memories and associations with past decades and childhood [81].

In the path from nostalgia inclination to nostalgic affection, nostalgia inclination had a significant positive impact on both historical nostalgia (0.447) and personal nostalgia (0.352). Individuals with a higher nostalgia inclination were more likely to experience nostalgic affection. Nostalgia inclination reflects the difficulty level of individuals' nostalgic experiences and their "attitudes towards the past" [82]. It was influenced by personal characteristics. This study analyzed the differences in nostalgia inclination among different genders, ages, education levels, and rural experiences using independent sample *t*-tests and one-way ANOVA analysis and found that women had a stronger nostalgia inclination than men (Table 9). This may be related to women's stronger affective needs and perception abilities. Furthermore, it can be observed that there was little difference in nostalgia inclination among different age groups (F = 2.214, *p* = 0.095), with slightly higher nostalgia inclination among those over 70 years old. This is similar to the current doubts about the absolute relationship between nostalgia and age [83]. There were no significant differences in nostalgia inclination among different educational levels (F = 0.156, *p* = 0.926) and elderly groups with or without rural experience (t = 1.497, sig. = 0.139). This suggested that nostalgia is a "naturally occurring phenomenon" that has become integrated into residents' daily lives and cognitive patterns, regardless of their upbringing environment.

**Table 9.** Differences in nostalgia inclination between different genders.

|  | Male (M ± SD) | Female (M ± SD) | t | Sig. |
|---|---|---|---|---|
| nostalgia inclination | 3.85 ± 0.81 | 4.28 ± 0.82 | −2.190 | 0.032 |

(2) In the path from nostalgic affection to place attachment, the paths from personal nostalgia to place dependency, historical nostalgia to place dependency, and historical nostalgia to place identification were supported, while the path from personal nostalgia to place identification was not established. This pointed out that the memories evoked by nostalgia in older adults pay more attention to the connections between people, places, and history rather than reaching the level of "affective identification". Nostalgia is a widespread phenomenon in contemporary society and is not something that occurs out of thin air. It is rooted in profound local cultural backgrounds, and nostalgia and place are closely linked and interconnected [84,85]. Chase et al. pointed out that nostalgia can only occur in a cultural environment of linear time (i.e., history), and it has a "sense of lacking something in the present". Ruins and artifacts from the past become material existences for nostalgia [86]. It can be seen that the essence of nostalgia is a sentimental state constructed from nostalgia for past eras, and it attempts to recreate and reproduce everything from the past through symbols and representations of the past [87]. Boym believed that nostalgia is both "restorative" (reconstruction of the past) and "reflective" [88]. Nostalgia includes memory, but it creatively combines time memory with emotional experience, providing a strong explanatory power for landscape experiences [89]. Currently, nostalgia as a commercial tactic has appeared in historical streets, cultural heritage sites, museums, coffee shops, restaurants, and other places. Nostalgic emotions can also be used for place construction and place identification. This research once again supported the argument that nostalgia and place are closely connected.

(3) In the paths of "landscape perception → environmental restorative effect" and "place identity → environmental restorative effect", the former is a subjective evaluation of the environment by individuals, which can enhance the restorative effects of the environment. This is consistent with existing research findings [90]. The correlation ranking of various aspects of landscape perception was as follows: L6 cultural characteristics of symbolic landscape (0.922) > L7 affective enhancement of soundscape (0.915) > L3 affective response to landscape sculptures (0.825) > L4 color configuration of plant landscape (0.799) > L8 atmospheric rendering of olfactory landscape (0.723) > L1 visual beautification of landscape topography (0.717). The unique cultural temperament of landscapes, historical conveyance of sculptures, visual impact of plants, harmonious variations of topography, and auxiliary embellishment of soundscapes and scent landscapes can enhance the satisfaction of older adults with their surrounding environment. This can help them escape from mundane daily routines, transcend worldly matters, achieve the desired match between expectations and the environment, and improve their mood, cognitive restoration, fatigue elimination, and revitalization. Among them, except for the visual aspect of landscape construction, it should be emphasized that the acoustic environment and odor environment play an important role in it. Many studies on nostalgia therapy involved music intervention [91] and odor creation [44], which can provide some evidence for this outcome. The path of the latter has also been confirmed by relevant research [16]. The restorative effects of the environment on individuals were not only determined by natural characteristics but also by the tolerance of local attachment. Another study suggested that only local identity, coupled with familiarity and preference, can play a significant positive role in assessing recovery potential. The elderly participants in this survey all had a certain level of familiarity with the sample park, as this research conclusion was also indirectly verified [92].

*5.2. Discussion on the Mediating Role of "Nostalgia Affection" in Environmental Restorative Effects*

In the mediation effect analysis, the study verified the existence of three paths: "landscape perception → nostalgic affection → environmental restorative effects", "landscape

perception → place attachment → environmental restorative effects", and "landscape perception → nostalgic affection → place attachment → environmental restorative effects". In terms of spatial cognition modes and approaches, they can be divided into "scientific cognition" and "experiential cognition", as well as "experiential spatial cognition" and "constructive spatial cognition". Scientific and experiential cognition have objectivity and exclusivity and are the direct result of abstracting and rationalizing space through the senses. Experiential and constructive cognition requires the emotional participation of cognitive subjects [93]. Compared with young people, older adults have formed different preference characteristics and psychological perception processes under the interweaving of these two cognitive styles. Additionally, prolonged experience and accumulated knowledge make subjective cognition more prominent. From the perspective of environmental psychology, the process of urban park restorative effects is a gradual psychological perceptual process, especially in environments with place attachment, which undergoes the psychological process of "environmental cognitive evaluation → affective response → emotional activation" [94]. First, if the park environment where older adults are located meets their preference needs and triggers their historical and individual nostalgia for the environment, then the environment can enhance their positive emotions in the park. Second, if the park environment is preferred by older adults, it will trigger their adaptive dependence on the environment, thereby enhancing their sense of place identity and the restorative effect of the environment on them. Third, when older adults are in high-quality landscape environments, the park can enhance their attachment to the environment by triggering nostalgic affection, further promoting physical and mental health restoration.

*5.3. Discussion on Landscape Optimization Strategies under the Goal of Healthy Aging*

To further discuss how to implement it in planning practice, the relationships between landscape type preference, nostalgia inclination, and landscape perception were excavated. On the one hand, as mentioned by landscape type preference, older adults' perception of the landscape in the park is the result of evaluating the affective atmosphere conveyed by various elements, such as the visual environment, sound environment, scent environment, etc. It includes factors such as sculptures, plants, topography, and cultural representations. The highest average preference among several types of landscape was plant landscape and natural landscape, followed by ancient architecture, cultural landscapes, and rural landscapes. Although the overall preference for natural landscapes was high, there were still some older adults with low demand for natural landscapes [83]. This may be on account of the fact that aging is a process of regression, and cognitive functions in older adults go through cognitive developmental stages in reverse order [95]. The individual aging of older adults includes a decline in biological physical and mental functions, as well as a decline in sensory and perceptual functions and the individual's ability to obtain information from their surroundings. Compared with artificial environments, natural landscapes have a higher degree of unknownness. Their "weak" cognitive characteristics thereby lead to a low preference for natural landscapes. It should be noted that this is not contradictory to the fact that natural landscapes are beneficial for the mental health recovery of older adults. In practical applications, these two should be considered together. Regarding ancient architecture and cultural landscape, older adults with a higher education level—those who graduated from junior high school—had a higher preference for cultural landscape (F = 5.628, $p = 0.002$), but their preference for ancient architecture was not significant (F = 1.802, $p = 0.155$). This illustrated that older adults who have received more education may pay more attention to the cultural value of landscapes, such as historical exhibition halls in parks. Although ancient architecture has a sense of the times, it is widely used in Chinese gardens and does not produce significant differences. There was no significant difference in rural landscape preferences among different rural living conditions (t = 1.013, sig. = 0.315), which was related to the reality that older adults with or without rural living experience have extreme yearning or uncomfortable attitudes towards the countryside. Older adults who maintain a longing for rural landscapes believe that rural landscapes are

closer to nature, away from the hustle and bustle of the city, and could evoke a positive mental state in their youth. Others who express discomfort with rural landscapes may have left a negative impression on rural landscapes due to poor infrastructure and living conditions in the past. Thus, the preference for rural landscapes presented a two-stage differentiation.

On the other hand, the correlation between landscape preference and nostalgia inclination had also been established (Table 10), and the results showed that among these five landscape types, the most correlated one was the rural landscape. This confirmed that rural nature is the core attraction of rural tourism [96] and was consistent with the induction of "homesickness" and "nostalgia" led by existing rural and nostalgic tourism [40,97,98]. Plant landscapes and natural landscapes follow closely, which is related to the biologically-based theory of common landscape preferences in humans [99,100], such as a higher preference for natural landscapes over artificial ones [101,102], especially a strong preference for "deep ecological" landscapes that make people feel the greatness and power of nature [103].

**Table 10.** Correlation analysis between nostalgia inclination and landscape preference.

|  | Natural Landscape | Traditional Architecture | Plant Landscape | Humanistic Landscape | Rural Landscape |
|---|---|---|---|---|---|
| nostalgia inclination | 0.401 ** | 0.326 ** | 0.408 ** | 0.312 ** | 0.614 ** |

** $p < 0.05$.

To further understand the relationship between older adults' landscape preference and landscape perception, a correlation analysis was conducted between the two as well (Table 11). It can be revealed that overall, for natural landscapes such as mountains, rivers, and plants, which are centered around natural elements, stimuli from different sensory dimensions are more important, including the creation of plant color, sound, and odor landscapes. For ancient architecture and cultural landscapes centered around human history, the symbolization and landmarks of landscapes are more important. For rural landscapes that are familiar to most elderly groups, their needs cover various aspects. These significant elements can be used as a lever to enhance landscape preference and nostalgic triggers, providing guidance and ideas for the implementation of park optimization practices.

**Table 11.** Correlation analysis between landscape preference and landscape perception.

|  | L1 | L3 | L4 | L6 | L7 | L8 |
|---|---|---|---|---|---|---|
| natural landscape | 0.138 | 0.217 * | 0.247 ** | 0.117 | 0.102 * | 0.209 * |
| traditional architecture | 0.116 | 0.179 | 0.130 | 0.125 * | 0.043 | 0.167 |
| plant landscape | 0.130 | 0.219 * | 0.253 ** | 0.124 | 0.103 | 0.193 * |
| humanistic landscape | 0.152 | 0.228 * | 0.129 | 0.155 * | 0.143 * | 0.287 * |
| rural landscape | 0.244 * | 0.456 ** | 0.359 ** | 0.365 ** | 0.356 ** | 0.445 ** |

* $p < 0.1$, ** $p < 0.05$.

In summary, in order to achieve the dual goals of improving park homogenization and improving elderly health, this study introduced the "nostalgia" sentiment and proposed corresponding planning and design strategies from two aspects: shaping landscape types and improving landscape quality. For one thing, in shaping landscape types, it is necessary to coordinate the current situation and adapt to local conditions. The overall types of landscapes can be divided into "naturalness", "historicalness", and "ruralness". "Naturalness" is the most fundamental feature of a park, carrying out its basic recreational and ecological goals. If "nostalgia" is involved in the goal of elderly health recovery, it can generate landscape types of "naturalness + historicalness" and "naturalness + ruralness" deprived of this. If the park is located in a remote area and has a large scale, its "historicity" can be explored first, and group memory can be used to trigger nostalgia, emphasizing the creation of nostalgic situations and atmospheres. The "Educated Youth Park" in Shaoxing City provides a successful case study. From the 1950s to the late 1970s, a large number of

urban educated youth left the city on a large scale and settled in the vast rural areas to participate in labor. This is known as the "educated youth going to the mountains and countryside" movement, which has affected over 20 million families. The park is set against the backdrop of ancient trees, and various historical landmarks such as giant hot water kettles, enamel tea cups, wheels, and other sculptures connect to form a coherent narrative landscape, making visitors who have shared experiences feel like they have traveled back to the past era. At present, the park has become a popular check-in spot for Internet celebrities and is well-received by the middle-aged and elderly population. If it is not possible to explore the history, one can consider integrating the concept of "ruralness" and combining ecological and cultural protection to consolidate industrial localization and sustainable development. In addition to the mentioned educated youth park, another expression is "Country Park", and a representative case is "My Country Park" in Chengdu. Its core is based on its own ecological pattern, maximizing the protection of the original ecology, retaining indigenous people, original housing, and original property rights, and using raw materials; By relying on natural mountains and water such as fish ponds, vegetable gardens, fruit trees, and forest trays, the main landscape axis is created, while secondary landscape axes are formed along the main roads. In short, it means beautifying agricultural and forestry landscapes, enhancing the background of agricultural landscapes through land art and landscape layout, and creating a creative and poetic rural environment. Meanwhile, internet technology can be combined to expand promotional channels and form models of "history + nostalgia" and "rurality + nostalgia".

For another, in terms of improving landscape quality, full attention should be paid to the optimization of landscape elements and features, especially in the limited scale and existing park renovation. First and foremost, symbolic landscapes infused with cultural regional characteristics can be shaped, and the uniqueness of parks can be explored through cultural precipitation, even creating brand effects. Currently, such attempts are increasing. Case in point: the improvement of natural landscapes can use landscape sketches, plants, soundscapes, and odorous landscapes as nostalgic triggers. Ancient architecture should take symbolic landscapes as its starting point; the plant landscape ought to start with the creation of landscape sketches, plants themselves, and odors; the cultural landscape is supposed to emphasize landscape sketches, symbolic landscapes, and dimensions of sound and odor perception; and the rural landscape had better integrate various elements and dimensions, especially the enhancement of odor landscapes and landscape sketches. After that, it is worth emphasizing that creating a good sound scene and integrating natural or artificial sound scenes, such as playing nostalgic songs and music, can better enhance the atmosphere and form more significant emotional feedback. Furthermore, it can be combined with rural experience methods to increase the "local flavor", such as micro-picking and gardening projects, to create a good-smelling landscape. Moreover, the configuration of plants, sculptures, and terrain should be coordinated to create a coherent and integrated narrative landscape structure.

## 6. Conclusions

Based upon the above results and discussions, the entire psychological process can be understood as the formation of different nostalgia inclinations and landscape preferences among older adults influenced by age, gender, and education level, etc. Older adults with a positive attitude towards the past will promote the occurrence of individual nostalgia and historical nostalgia among them. Affected by environmental quality, older adults in the environment further formed different landscape perception evaluations. At the same time, landscape perception can promote their local dependence and identity, and place dependence can also enhance their sense of place identity. Furthermore, place dependence and identification will enhance the restorative effects of the environment. It can be summarized as follows: in the first place, "nostalgia" played an important role in the process of environmental restorative effects, with "nostalgia affection" being an important mediating factor; in the second place, the pathways of "landscape perception →

nostalgic affection → environmental restorative effects", "landscape perception → place attachment → environmental restorative effects", and "landscape perception → nostalgic affection → place attachment → environmental restorative effects" all existed. In the third place, planning and design personnel can start with elderly individuals, consider landscape preferences and nostalgia tendencies, and explore optimization strategies. The optimization strategies provided in this article mainly involved two aspects: shaping landscape types and improving landscape quality, which also corresponded to the difficulties and challenges of current park green space planning. On the one hand, the special planning and overall planning of urban green spaces should form an interaction to optimize the living space environment and promote healthy and sustainable urban development. This requires planning to strengthen pertinence and deepen the exploration of characteristics that combine urban green space planning with the development history, urban nature, natural environment, and plant species distribution of the city. On the other hand, for the planning or renovation of specific projects, various elements should be reasonably arranged and configured, their own cultural characteristics should be explored, multi-sensory experiences should be built, emotional triggering effects should be integrated, and group memory should be awakened. For instance, Changchun can use the culture of "Automobile City" and "Film City" as the foundation to renovate existing parks and agricultural ecological landscape parks and create scenic spots with nostalgic themes or rural experiences. While coordinating to meet the leisure and entertainment needs of residents, it can achieve social benefits and healthy aging.

Due to the selection of samples and the difficulty of the research, this research has not conducted extensive exploration in certain aspects. Nevertheless, it can still provide new directions and ideas for exploring restorative environments. In the future, research teams can further explore the affective experiences and natural experiments of different groups, providing theoretical support and implementation strategies for sociology, psychology, and planning disciplines.

**Author Contributions:** Conceptualization, T.Y.; methodology, T.Y.; software, T.Y.; validation, T.Y.; formal analysis, T.Y.; investigation, T.Y.; resources, T.Y.; data curation, T.Y.; writing—original draft preparation, T.Y.; writing—review and editing, T.Y. and H.L.; visualization, T.Y.; supervision, T.Y., H.L. and Q.Y.; project administration, H.L. and Q.Y.; funding acquisition, H.L. and Q.Y. All authors have read and agreed to the published version of the manuscript.

**Funding:** This research was funded by National Natural Science Foundation of China (51978192).

**Data Availability Statement:** Not applicable.

**Acknowledgments:** We would like to thank the editor and the anonymous reviewers for their time and feedback, which substantially improved this work.

**Conflicts of Interest:** The authors declare no conflict of interest.

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
