# Peer review of "The Role of “Nostalgia” in Environmental Restorative Effects from the Perspective of Healthy Aging: Taking Changchun Parks as an Example"

_land, doi:10.3390/land12091817_

Round 1

Reviewer 1 Report

Thank you for the opportunity to review this manuscript. Based on a current topic of the important influence of health and well-being in urban parks, combined with structural equation modeling,this study discusses the important impact of aging people's "nostalgia" emotion on the restorative effect of park environment. I understood and certainly this work has practical implications in restorative environment construction in park space, especially for the elderly. The structure of this manuscript is clear and concise to understand the results. Nevertheless, there are some deficiencies in research methods and contents.

Here I have several questions or suggestions in order to improve readability and clarity and are reported below:

1)       Introduction. In the first sentence of the introduction, the author expounds the current situation of aging in China. I suggest providing detailed data as the support of the argument, and the aging situation of China and the cities in the study area, so as to facilitate the understanding of international readers.

2)       Introduction. The author mentions “healthy aging”, what is the concept and content, it needs to be elaborated and supplemented with relevant references.

3)       Introduction. The author pays attention to the research of "nostalgia" emotion of the elderly, but the current situation of the relationship between "nostalgia" emotion and park landscape environment, and the influence of "nostalgia" emotion and restorative benefit is not clear enough. This is the main basis for the establishment of the author's research hypothesis, and it is recommended to add more relevant literature to support.

4)       Materials and Methods. In this part, the evaluation dimension and specific content of landscape perception are not clear enough. For example, what specific indicators need to be reported in the evaluation of soundscape and olfactory landscape.

5)       Materials and Methods. In assessing the benefits of environmental restoration, the authors focused on mental health. Whether the selection of dimension indicators of mental health takes into account the indicators directly related to emotions may be more suitable for this study than the broad psychological indicators. For example, among the four existing indicators, R1 eliminates fatigue and R4 restores attention, which may not be a major problem for emotional health in elderly.

6)       Materials and Methods. I suggest that the author should elaborate on the process of data collection, the time when 210 questionnaires were collected and the characteristics of specific sites in the park. According to the site photos provided in table 2, there are obvious differences in the site characteristics of the four parks. The author does not report whether the specific environment of the park is investigated. It should be reminded that there are differences in the impact of water features, cultural landscapes and green environment on human health. In addition, as a questionnaire study, I think the sample size of this study is too small.

7)       Discussion. In this study, the specific landscape characteristics and environmental replacement of the investigation site are not clear enough, which is an important limitation affecting path construction. The extensive research results, such as the positive impact of scenery on human health, are difficult to guide practical application.

 Moderate editing of English language required

Reviewer 2 Report

Dear Authors,

The paper displays a reasonably clear structure and adopts an intriguing perspective on healthy ageing and the significance of restorative environments, particularly in the context of parks. However, there are specific suggestions that could help refine and enhance the paper's quality:

In my view, the introduction feels overly dense with an excess of information. Simplifying certain phrases and breaking down the information into more digestible segments might be beneficial. Consider revising sentences such as "China has entered an aging society and the rapid growth rate cannot be ignored."

When referencing studies or theories, it would be beneficial to have specific examples or key findings from these studies. For instance, instead of merely stating "Attention Restoration Theory and Stress Reduction Theory," I suggest incorporating a brief sentence on what these studies unveiled or how they directly relate to the subject of ageing and parks.

Furthermore, when introducing the term "nostalgia," a concise definition or description of its implication in this context, beyond its acknowledgment in modern psychology, would be beneficial.

The section highlighting issues with existing research might benefit from expansion to provide a clearer idea of the specific knowledge gaps. For instance, when noting that current research has led to a "significant homogenisation of parks," please offer tangible examples of what this entails and why it's problematic.

Lastly, by the end of the introduction, it would be helpful to clearly reiterate the study's objectives and aims. For example: "With this study, we aim to specifically understand how nostalgia, as a key emotion experienced by older adults, plays a role in the restorative effects of parks and how this can inform the planning and design of more effective and health-promoting spaces."

I believe a more comprehensive contextualisation is needed, offering a broader perspective on why considering the emotional and physical needs of older adults in urban planning is crucial. One might discuss how the ageing population will impact urban infrastructures and systems in the coming decades.

Regarding the hypothesised assertions, I suggest listing the hypotheses and providing a brief justification and description following each one. The importance of these hypotheses to the study needs emphasis. For example:

h1. Landscape perception can positively influence historical nostalgia.

Justification: Elderly individuals who have lived in rural environments might resonate with landscapes resembling such settings, invoking nostalgic emotions.

Concerning materials and methods:

I recommend clearer articulation regarding latent variables. When defining the 7 latent variables, a brief definition or description of each would help the reader better grasp their meaning and relevance in the survey context. Additionally, while some latent variables have a detailed item description (e.g., landscape perception), others like "inclination towards nostalgia" only mention being based on prior scales. It would be useful to specify the total number of items for each latent variable.

Concerning data collection, although it's mentioned that Changchun City was selected due to its serious ageing situation, offering more context about why it's an ideal site for investigating landscape preferences and the mentioned latent variables would be beneficial. It's mentioned that older adults visiting these parks for the first time were excluded. What's the rationale behind this exclusion? It's noted that adjustments were made to observed variables after a pilot survey. Elaborating on what type of feedback or pilot survey results led to the removal of specific items would be beneficial.

Referring to statistical methods, while the choice of SEM is briefly justified, delving deeper into why it's the most suitable method for this specific study would be useful. Before conducting an SEM analysis, it's vital to ensure the data meets certain assumptions. Describing any prior validation steps or assumption checks would be beneficial.

Pertaining to the Results section:

The Results section's structure is generally lucid, but I suggest enhancing the text's flow for increased reader accessibility. Phrases such as "Essentially, in the evaluation of environmental restorative effects..." and "Then, parks had better evaluation of the emotional improvement..." appear somewhat disjointed and could benefit from clearer transitions.

In the "4.1. Evaluation Results of Various Dimensions" section, it's mentioned that average scores ranged between "agree to very agree." More context about the used scale would be useful. For instance, was the scale from 1 to 5 or 1 to 7? This would help readers understand the scores' significance.

For assertions like "indicating that the health restorative effect of the sample park on older adults was positive and beyond doubt", I suggest including a justification or reference to back this claim. Is there prior literature that supports this range as positive?

It would be valuable for the authors to provide further details about the study's sample, especially in the results section.

In "Table 5. The path results of nostalgia", it's beneficial to have provided the p-Value, but it would be even more advantageous if the author distinctly defined the significance level in use (e.g., p < 0.05, p < 0.01).

Including a brief discussion or interpretation after each results sub-section to provide context and meaning would be valuable.

Terms such as "landscape perception", "place attachment", and "nostalgic affection" should be clearly defined at the article's onset to ensure reader comprehension.

When discussing landscape types, offering specific examples of parks or places that exemplify "naturalness + historicalness" or "naturalness + ruralness" might be beneficial.

The manuscript frequently alludes to the implications of the research on planning and design, but I believe connecting more explicitly with existing theoretical frameworks or challenges in urban planning would be beneficial.

Round 2

Reviewer 1 Report

Dear author,

I think this manuscript has been properly revised to meet the requirements for publication. I believe this study will promote the research and practice of healthy landscape design. Congratulations! We also look forward to new breakthroughs in this field!

Reviewer 2 Report

I appreciate the careful consideration and incorporation of my suggestions by the authors. It is evident that the manuscript has been strengthened and refined based on the feedback provided.